# Designing a Multi-Epitope Subunit Vaccine against VP1 Major Coat Protein of JC Polyomavirus

**DOI:** 10.3390/vaccines11071182

**Published:** 2023-06-30

**Authors:** Sukhada Kanse, Mehak Khandelwal, Rajan Kumar Pandey, Manoj Khokhar, Neetin Desai, Bajarang Vasant Kumbhar

**Affiliations:** 1Department of Biological Sciences, Sunandan Divatia School of Science, NMIMS (Deemed to be) University, Vile Parle (West), Mumbai 400056, Maharashtra, Indianeetindesai@gmail.com (N.D.); 2Department of Medical Biochemistry and Biophysics, Karolinska Institute, 17177 Stockholm, Sweden; 3Department of Biochemistry, All India Institute of Medical Sciences (AIIMS), Jodhpur 342005, Rajasthan, India

**Keywords:** progressive multifocal encephalopathy (PML), JC polyomavirus, VP1 protein, immuno-informatics, molecular dynamics simulation

## Abstract

The JC polyomavirus virus (JCPyV) affects more than 80% of the human population in their early life stage. It mainly affects immunocompromised individuals where virus replication in oligodendrocytes and astrocytes may lead to fatal progressive multifocal encephalopathy (PML). Virus protein 1 (VP1) is one of the major structural proteins of the viral capsid, responsible for keeping the virus alive in the gastrointestinal and urinary tracts. VP1 is often targeted for antiviral drug and vaccine development. Similarly, this study implied immune-informatics and molecular modeling methods to design a multi-epitope subunit vaccine targeting JCPyV. The VP1 protein epitopic sequences, which are highly conserved, were used to build the vaccine. This designed vaccine includes two adjuvants, five HTL epitopes, five CTL epitopes, and two BCL epitopes to stimulate cellular, humoral, and innate immune responses against the JCPyV. Furthermore, molecular dynamics simulation (100 ns) studies were used to examine the interaction and stability of the vaccine protein with TLR4. Trajectory analysis showed that the vaccine and TLR4 receptor form a stable complex. Overall, this study may contribute to the path of vaccine development against JCPyV.

## 1. Introduction

The central nervous system demyelinating disease known as progressive multifocal leukoencephalopathy (PML) is caused due to the John Cunningham virus, also known as Human Polyomavirus-2 [1]. The mode of transmission of JC polyomavirus (JCPyV) is given in Figure 1. It is a hemagglutinating, widespread, and species-specific disease condition. The virus has a seroprevalence between 40 and 60%, whereas in children, seroprevalence is only 10–20% and significantly rises in the following years [2]. The virus can be classified into subtypes depending on how widely it has spread among diverse human populations. Types 1 and 4 typically affect Europeans, types 3 and 6 affect the African population, and type 7A typically affects South-East Asians [3].

JCV is a non-enveloped virus comprised of 72 viral capsomeres with icosahedral symmetry (Figure 2A,B), each containing a circular dsDNA genome. The viral genome has two coding domains and is approximately 5000 base pairs long [5]. For example, the first viral gene encodes large and small tumor antigens. The second region, in contrast, encodes for the three main structural proteins VP1, VP2, and VP3, as well as an auxiliary regulatory protein known as agnoprotein [6]. The early and late viral genes are separated by a regulatory non-coding control region (NCCR). The NCCR contains transcription factor-binding sites, host cell-specific DNA, the replication origin, and regulatory areas focusing on early and late transcription [7].

Numerous studies have demonstrated the significance of the primary capsid protein VP1 in promoting JCV attachment to host cell receptors and starting a viral infection [8,9,10]. The oligosaccharide lactoseries tetrasaccharide C (LSTc), composed of 2,6-linked NeuNAc, is the only one with which JCV-VP1 interacts. Thus, LSTc functions as a JCV receptor [11]. It has been discovered that VP1 also interacts with the 5-HT2AR serotonin receptor. However, it is unclear how the 5-HT2AR functions in infections [12]. The VP1 protein also facilitates viral assembly inside the host cell nucleus [13]. Because of this, VP1 is a crucial target in creating vaccines and drugs to treat JCV infection.

The SARS-CoV-2 outbreak [14] demonstrated the importance of vaccination when the entire world was waiting to develop a potent vaccine to end the pandemic. Moreover, recent events have highlighted the importance of vaccination. Earlier, the immuno-informatics approach was used to develop the vaccine against various viral infections [15,16,17,18]. Recently, the synthetic binding proteins database was developed, which could lay a solid foundation for future research, diagnosis, and therapy development [19]. We have attempted to design a suitable and stable vaccine for preventing JCPyV-induced PML. Using an innovative immuno-informatics method, VP1 protein was targeted to construct a subunit vaccine. Various web servers were utilized to predict different epitopes for the vaccine construction, and adjuvants and linkers were included. It was determined whether the developed vaccine had strong antigenic and non-allergenic qualities that could stimulate an effective immune response against the JCPyV. The vaccine binding mechanism and stability with the TLR4 receptor were further explored using molecular docking and molecular dynamics simulations. The immunological simulation proves the vaccine’s durability and antibody generation.

## 2. Methodology

### 2.1. VP1 Protein Sequence Retrieval and Physicochemical Analysis

The VP1 capsid protein’s sequence (UniProt ID: P03089) was retrieved from the UniProt database [20]. The physical and chemical features of the VP1 proteins were ascertained using the ExPASy-ProtParam tool [21]. The VaxiJen v2.0 server was employed to determine the protein antigenicity [22]. This server performs antigenicity analysis using the physiochemical characteristics of the protein. The NCBI-BLAST program determined the homology between the VP1 coat protein and other animals, primarily humans. Their sequences were further matched, and the phylogenetic tree was built using the Neighbor-Joining method through the MEGA11 software [23] with default settings and 1000 bootstrap replicators (Figure 2C).

### 2.2. B Cell Epitopes Prediction

The B cell epitopes were predicted by using the Immune Epitope Database (IEDB) server. To anticipate the epitopes, the Kolaskar and Tongaonkar approach was employed [24]. It uses amino acids’ physical and chemical characteristics to predict the B cell epitopes. Hence, the FASTA format VP1 main capsid protein sequence was provided with a threshold value of 1.022. Further submissions of the predicted B cell epitopes to the ToxinPred service were made to verify their non-toxicity [25]. VaxiJen v 2.0 [22] server was used to predict the antigenicity of the epitopes, and the AllerTOP v.2.0 server [26] to sort the non-allergic epitopes.

### 2.3. Helper T Lymphocyte (HTL) Epitopes Prediction

Antigen-presenting cells express MHC-II molecules on their surface, presenting antigenic peptides to the T cell receptor, which coordinates the outcome of the host’s immune response. The MHC-II specific epitopes for the VP1 capsid protein were identified using IEDB MHC-II tool [24]. This tool uses the NN-align-2.3 (netMHCII-2.3) algorithm to predict the same using an Artificial Neural Network (ANN) [27]. The predicted scores are given in IC50 values (nM) and as a percentile rank. The sorted peptides were further scanned for antigenicity using VaxiJen v 2.0 [22], allergenicity using AllerTOP v.2.0 [28], and toxicity using ToxinPred server [25].

### 2.4. Cytotoxic T-Lymphocyte (CTL) Epitopes Prediction

The prediction of the CTL epitopes aims to identify the key peptides that stimulate the CD8^+^ T cells. The Immune Epitope Database (IEDB) MHC Class I tool was used to identify the CTL epitopes in accordance with the ANN [24,25,26,27]. The neural network is the most significant algorithm to sort the peptides that have strong affinity to HLA molecules, demonstrating that high-binding peptides are the main source of the higher-order sequence correlation signal. The three essential parameters that were concentrated on while predicting the epitopes were MHC binding affinity, proteasome cleavage, and TAP transports. The results obtained were further scanned for antigenicity, allergenicity, and toxicity.

### 2.5. Computational Construction of Vaccine and Its Physiochemical Analysis

Here, the B cell epitope, HTL epitope, and CTL epitopes were assembled in a sequence-wise manner. Different connecting linkers were used to connect these different epitopes. These epitopes were linked by KK, GPGPG, and AAY linkers, respectively [6]. To increase the vaccine’s effectiveness and immunogenicity, the N terminal of the construct was connected with Human β defensin (total amino acids 45) and LL37 (total amino acids 37). They were linked together by the EAAAK linker. Human β defensin behaves as an antimicrobial agent as well as an immunomodulator [29]. LL37 is a cathelicidin family member and has antimicrobial activity [30]. The novel formulation of Human β defensin and LL37 as a ‘combination adjuvant’ generates high antigenicity and acts interdependently to produce a Th1-based adaptive immune response [31]. The physiochemical analysis of the vaccine was performed using the ExPASy-ProtParam tool [21]. Additionally, antigenicity and allergenicity prediction were applied to the sequence. Figure 3A depicts the final vaccine construct.

### 2.6. Vaccine 3D Structure Prediction and Validation

The three-dimensional vaccine construct provides valuable information for understanding and regulating biological functions. The GalaxyWEB protein structure prediction server [32] was used to build the structure of the vaccine. The server predicted the 3-D structure from the vaccine construct using template-based modeling (TBM) and refined the loop and the terminus regions. TBM is a method for predicting the structure of a protein by aligning its amino acid sequence with a known, similar protein structure (template) from a protein database. This approach assumes that proteins with similar sequences tend to have similar structures. TBM can be used to produce a 3D model of a vaccine construct that is intended to trigger an immunological response in the context of vaccine design. The vaccine structure was validated using a Ramachandran plot through PDBsum [33].

### 2.7. Interaction of Vaccine with TLR-4 Using Docking

The TLR4 signal pathway is crucial for triggering the innate immune response [34] and is further responsible for cytokine release [35]. Earlier studies demonstrated that innate immunity system sensors such as Toll-like receptor 4 (TLR4) recognize mouse polyomavirus (MPyV) and cause the production of interleukin 6 (IL-6) and other cytokines without preventing virus replication [36]. To investigate the binding mode of the vaccine construct with TLR4, a PatchDock server [37] was used. The crystal structure of TRL4 (3FXI.pdb) was retrieved from the protein database. Using the FireDock server, the PatchDock docking results were optimized [38]. The least binding energy vaccine construct docked with the receptor TLR4 was considered as an initial starting conformation for simulation.

### 2.8. Molecular Dynamic (MD) Simulation

To investigate the refined binding mode of the vaccine construct with TLR4, MD simulation was performed using Gromacs2021.3 [39]. Amber ff99-SB force field parameters were used for the simulation of the TLR4-Vaccine construct complex. The simulation systems were solvated using a water module, TIP3P, in a cubic periodic box of 10 Å. The necessary number of counter ions was added in order to neutralize the simulation system. Using the ‘Parmed tool’, the amber ‘topology’ and ‘co-ordinate’ files were converted into Gromacs companion ‘top’ and ‘gro’ files, similar to an earlier study [40]. The energy minimization was performed with the steepest descent (5000 steps) and the conjugate gradient (2000 steps) methods. To equilibrate all the systems, 1 ns NVT and 1 ns NPT simulations were performed. The equilibrated system was further used for the production MD simulation for 100 ns time steps, and the long-range electrostatic interactions were calculated using the particle mesh Ewald method with a cut-off distance of 1.0 nm, a Fourier spacing of 0.16 nm, and an interpolation order of 4. [41]. The simulated trajectories and snapshots were further analyzed and visualized using ‘gmx’ tools of Gromacs [42] and visualized by PyMol software [43].

### 2.9. Immune Simulation Response of Vaccine Protein

To investigate the immune response and immunogenicity of the vaccine protein, the C-IMMSIM server was used [44]. Immunogenicity refers to the ability of a vaccine to induce an immune response, specifically the production of antibodies that recognize and neutralize the target pathogen. The strength and quality of the immune response, or immunogenicity, is a key factor in determining the efficacy of a vaccine. A vaccine with high immunogenicity is more likely to protect against the disease it is designed to prevent. The were 1000 simulation steps, the simulation quantity was 10 μL, and the random seed was 12,345. C-IMMSIM predicts immunological interactions by using position-specific score matrices for peptide prediction that are built from machine learning approaches. Here, we predicted the interaction of vaccine candidates with immune receptors to boost humoral and cellular immune responses.

## 3. Results and Discussion

### 3.1. Sequence Retrieval and Analysis

The VP1 major capsid protein sequence was collected from the UniProt database (UniProt ID: P03089). The antigenic property of the VP1 protein sequence was observed to be 0.4042, which is equal to the threshold value of 0.4. The physicochemical analysis revealed that the protein contains 354 amino acids. The molecular weight was found to be 39.6 kDa, the isoelectric point was obtained as 5.79, and the instability index was calculated to be 33.79, which classifies that the protein is stable. With an aliphatic index of 75.88, the protein was shown to be stable throughout a wide range of temperatures. In mammalian reticulocytes (in vitro), yeast (in vivo), and *Escherichia coli* (in vivo), the estimated half-life was 30 h, 20 h, and 10 h respectively.

Furthermore, the phylogenetic analysis study was performed to check the evolutionary relationship with polyomavirus (Figure 2C). A less branching pattern indicated minimal evolutions, and the phylogenetic analysis demonstrated that VP1 coat protein clustered together in a single clade, having the greatest common ancestry. Therefore, the vaccine designed against one strain can be used for all polyomavirus strains.

#### 3.1.1. B Cell Epitope Prediction

The B cell epitopes can induce humoral immunity, as they are recognized by the B cell receptors. In total, 14 epitopes were predicted by the IEDB B cell epitope prediction server. Further testing was undertaken using the antigenicity, allergenicity, and toxicity on these epitopes. The epitopes with the highest antigenic score, non-allergenic and non-toxic epitopes, were selected for the vaccine construct and are listed in Table 1. These vaccine epitopes are crucial for the immunological response.

#### 3.1.2. HTL Epitope Prediction

The HTL epitopes induce a CD4+ helper response, which aids in the expansion of protective CD8+ T cell memory and the activation of B cells for the production of antibodies. Hence, the IEDB MHC Class II epitope prediction server was used [24], and it predicted a total of 9047 HTL epitopes. The top five epitopes, which were antigenic, non-allergic, and non-toxic, were chosen to design the vaccine construct and are listed in Figure 3A and Table 2.

#### 3.1.3. CTL Epitope Prediction

CD8+ T cells play a crucial role in the development of long-term immunity and elimination of the circulating virus and virus-infected cells. A total of 16,606 CTL epitopes were predicted by the IEDB MHC Class I epitope prediction tool. The top five epitopes, which were antigenic, non-allergic, and non-toxic, were chosen to design the vaccine construct and are listed in Table 3.

#### 3.1.4. Design of the Linear Vaccine Construct and Physiochemical Analysis

The vaccine protein construct consists of 284 amino acids with a molecular weight of 31.39 kDa, as shown in Figure 3A. The protein is stable over a large temperature range, according to the instability index calculation, which was found to be 32.67. The vaccine construct was also found to be antigenic, non-allergenic, and non-toxic.

#### 3.1.5. Vaccine Tertiary Structure Prediction and Validation

The GalaxyWEB server [32] was used to predict the three-dimensional structure of the vaccine construct, as shown in Figure 3B,C. The three-dimensional vaccine model was validated using the Ramachandran plot generated using the PDBSum server, as shown in Figure 3D. It showed that 97% of residues are present in the most favored region and additionally allowed region, whereas only 0.9% were present in the generously allowed region and 2.2% in the disallowed region. The stereochemical quality of the vaccine construct was found to be good, and hence was further used for docking and MD simulation study.

#### 3.1.6. Binding Mode of Vaccine Construct with TLR-4

To explore the binding mode of the vaccine construct with the TLR4 receptor, docking was performed using the PatchDock server [37]. The 10 lowest energy docked outputs were selected and submitted to FireDock for further refinement [38]. The PIMA server [45] was used to understand the vaccine and TRL4 complex stability; the complex is stabilized by the electrostatic (−12.7055 kJ/mol) and van der Waals energy (−95.8678 kJ/Mol), and the total stabilizing energy is −108.5733 kJ/Mol. The FireDock output with the least global energy value of vaccine-TRL4 (Figure 4A), was considered as a starting structure for simulation. The TLR4 and vaccine complex is stabilized by mainly hydrogen bonding and non-bonding interactions. The details of the residues involved in the TLR4-Vaccine complex interactions are shown in Figure 4B,C. Moreover, the complex forms hydrogen bonding interactions between TLR4 (Chain-B) Gln81, His103, Ser127, and Glu286 with Vaccine (Chain-C) Ile262, Glu260, Thr259, and Leu210, respectively. The TLR4-Vaccine complex is also stabilized by the non-bonded interaction as shown in Figure 4B,C.

### 3.2. Molecular Dynamic Simulations

To explore the refined binding mode of the vaccine with the TRL4 receptor, MD simulation was run for 100 ns using Gromacs2021.3 [42]. The stability of the complex was checked by plotting the root mean square deviation (RMSD) of backbone atoms, as shown in Figure 5A. The RMSD plot reveals the stability of the TLR4-Vaccine complex, and RMSD was found to fluctuate below 1 nm (Figure 5A). Furthermore, the protein compactness was studied using the radius of gyration (Rg), as shown in Figure 5B. The Rg plot revealed the stable complex and compact state of the TLR4 and vaccine construct. Furthermore, the root mean square fluctuations were calculated for the vaccine and TLR4 chains A and B, as shown in Figure 5C,D.

The RMSF plot shows the movement of Cα atoms during the simulations (Figure 5C). The flexible region shows a higher RMSF, while the confined portion show a lower value of RMSF. In the RMSF plot, vaccine residues from regions 170 to 270 were found to have fewer fluctuations and were involved in the binding with the TLR4 receptor, as shown in Figure 5C. In addition, the TLR4 chain B residues from regions 0 to 250 were found to have fewer fluctuations and were involved in the binding with the vaccine construct, as shown in Figure 5C. Furthermore, the chain A residues from regions 300 to 400 were found to have fewer fluctuations and were involved in the binding with vaccine residues, as shown in Figure 4C and Figure 5C.

### 3.3. Immune System Simulation

To understand the immune response to the vaccine protein, the C-IMMSIM server was used. Here, the antigen and immunoglobulins, cytokine production, TH cell population, and B cell population parameters were studied and are shown in Figure 6. An increase in the IgM and IgG levels represents the main host response, as shown in Figure 6A. A subsequent reaction to the antigen is also shown by elevated levels of IgG1, IgG2, IgG, T cell, and B cell populations, as shown in Figure 6A–C. The increased expression of IFN-γ and IL is important for the inhibition of viral replication and cellular immunity (Figure 6B). Here, IFN-g, IL-4, IL-10, IL-12, and TGF-b shown expression compared to others (Figure 6B). The B cells and memory B cell formation were seen in the immuno-simulation, as shown in Figure 6D; this demonstrates the long-established immunological response induced by the vaccine construct. The results evaluation indicated that the vaccine could initiate the antigen-stimulated, more robust immune response and be safe. The immune simulation reveals the vaccine construct triggers the potential immune response and contributes to ensuring immunogenicity and stability, which are essential for vaccine development.

## 4. Discussion

The human JCPyV causes PML, a fatal demyelinating illness of the CNS. Only those with therapeutic immunomodulatory antibodies used to treat illness, including those with multiple sclerosis (MS), transplant recipients, and others are at risk. The high seroprevalence of JC virus and reduced incidence of viral infection is common and persists in an asymptomatic manner, gradually leading to viral reactivation causing PML [10,46].

The JCPyV archetype VP1 is the reason why the virus persists in the gastrointestinal and urinary tract. It results in virus infection through urine and feces. The JCPyV-PML mutations are located in the external loops of the VP1 protein [47]. JCPyVs with VP1 mutations result in a change in the receptor binding interaction and antibody escape, which can turn an asymptomatic infection in an organ-like kidney into a deadly brain disease [47]. There are still various unanswered queries about the disease mechanism of JC and PML. Currently, few treatment options are available for JC-induced PML, even though it is a rapidly progressing and devastating disease [10].

Previous studies have shown that a multi-epitope vaccine is an ideal approach for the treatment and prevention of viral diseases [48,49,50,51,52]. Different approaches are used to construct multi-subunit vaccines compared to the classical and single-subunit vaccines [15,16,17,18]. Designing a multi-subunit vaccine requires MHC-I and MHC-II restricted epitopes that can be recognized by T cell receptors (TCRs) of numerous clones by different T cell subsets. In contrast, B cell epitopes induce strong humoral immune responses to appropriate antigens for broadening the range of viral treatment, as well as to suitable adjuvants, and linkers for immunogenicity enhancement [51,52,53]. Thus, we focused on constructing a multi-epitope vaccine against PML caused by JCPyV using immuno-informatics and a molecular modeling approach.

To design an efficacious subunit vaccine, identifying conserved HTL and CTL epitopes was streamlined considerably using online computational tools. The selection of appropriate epitopes for the vaccine construct was made by passing several immune filters, including antigenicity, allergenicity, and toxicity, as per an earlier study [49]. The adequate epitopes for MHC-Class I and MHC-Class II were selected based on the highest VaxiJen score [22]. The epitopes for BCL were taken based on passing set parameters for selection. Here, Human β defensin (UniProt ID. Q5U7J2) was added as an adjuvant along with an EAAAK peptide linker known to increase bifunctional catalytic activity, an AAY peptide linker, and KK and GPGPG linkers, which bring the pH level near the physiological range, to construct the vaccine sequence [54]. The adjuvant, linkers, and epitopes used to construct the vaccine sequence were further analyzed on the set parameters and labeled as stable for future use. The physicochemical characteristics of the designed vaccine were analyzed to be within the threshold value, making it an adequate vaccine candidate [50,55]. Further, tertiary structure prediction of the resultant vaccine sequence was performed, and the Ramachandran plot was constructed for validation of the derived structure, and the stereo-chemical quality of the vaccine was found to be good (Figure 3D). The engagement of vaccines with target immune receptors is crucial for generating a persistent immune response. Therefore, the binding mode and interactions of vaccine protein and TLR4 were investigated using molecular docking and molecular dynamics simulation, as shown in Figure 4 and Figure 5. The simulation study reveals the stability of the vaccine construct with the TLR4 receptor. Furthermore, the immunologic response of a vaccine protein was analyzed using in silico simulations through the C-IMMSIM server [44]. This showed the immune response after injection by expressing different antibodies and stimulating a more robust immune response, as shown in Figure 6. Taken together, we propose a theoretical framework for the vaccine to combat JCPyV infections. We tried to address the implications of our study on the development of a JCPyV vaccine. Our multi-epitope subunit vaccine construct holds promise in tackling the challenges associated with progressive multifocal leukoencephalopathy (PML) and JCPyV infections. By targeting multiple viral epitopes, our vaccine aims to enhance immunogenicity and broaden the coverage of antigenic variants. This approach may reduce the risk of viral escape mutants and provide more comprehensive protection. Moving forward, further research, including preclinical and clinical evaluations, is necessary to determine the safety, immunogenicity, and efficacy of our vaccine, thus paving the way for its potential clinical application.

## 5. Conclusions

The JCV is a type of human polyomavirus that is a common cause of progressive multifocal leukoencephalopathy (PML), a disease that affects the brain’s white matter. PML is a rare but severe complication of immunosuppression, particularly in people with weakened immune systems due to other disease conditions such as HIV/AIDS or cancer. Currently, there is no vaccine available to prevent JC virus infection. The development of a vaccine against the JC virus is a complex challenge due to the virus’s ability to persist in the human body without causing symptoms. A JCV-induced PML diagnosis in immunocompromised and transplant patients is often non-responsive to existing treatment, and more than 50% of patients with JCV-induced PML will die without any treatment. The prevalence of JCV and PML will increase in immunocompromised and transplant patients. Therefore, our multi-epitope vaccination may provide protection against JCPyV, preventing viral replication and helping to develop effective strategies for PML.

### Limitation

This study further needs experimental validation, including the synthesis and in vitro and in vivo validations, to determine the vaccine’s immunogenicity. Antigenic variability is also a limitation. Many pathogens have the ability to rapidly evolve and change their epitopes, reducing the effectiveness of a multi-epitope vaccine.

## Figures and Tables

**Figure 1 vaccines-11-01182-f001:**
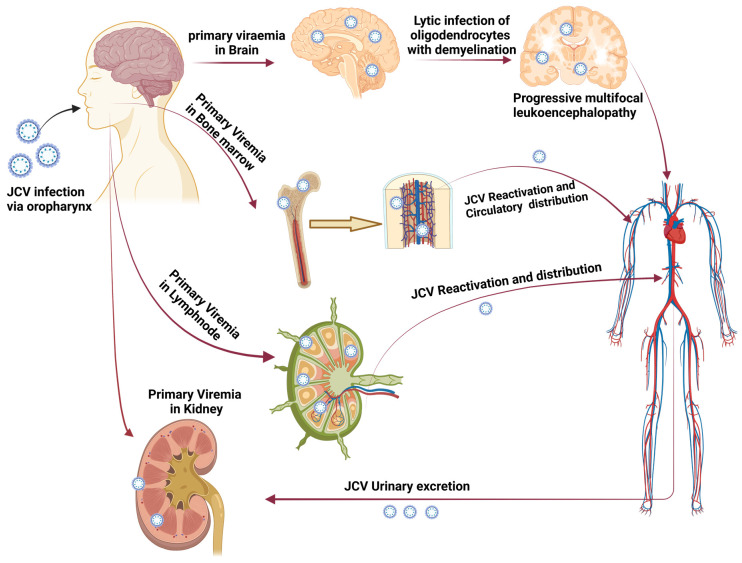
Transmission of JC polyomavirus. The figure shows how primary viremia is caused by virions entering the body through the tonsil epithelium, upper respiratory tract, and gastrointestinal system (mainly archetypal and infrequently neurotropic type). Further, it shows how viremia affects the kidney and other organs. JCV replication takes place in the kidney, and the JCV life cycle is finished by viruria caused by shedding from the apical face. Through bone marrow, JCV enters the brain hematogenously together with leukocytes. PML is indicated by the development of brain microlesions [4] (created using Biorender.com).

**Figure 2 vaccines-11-01182-f002:**
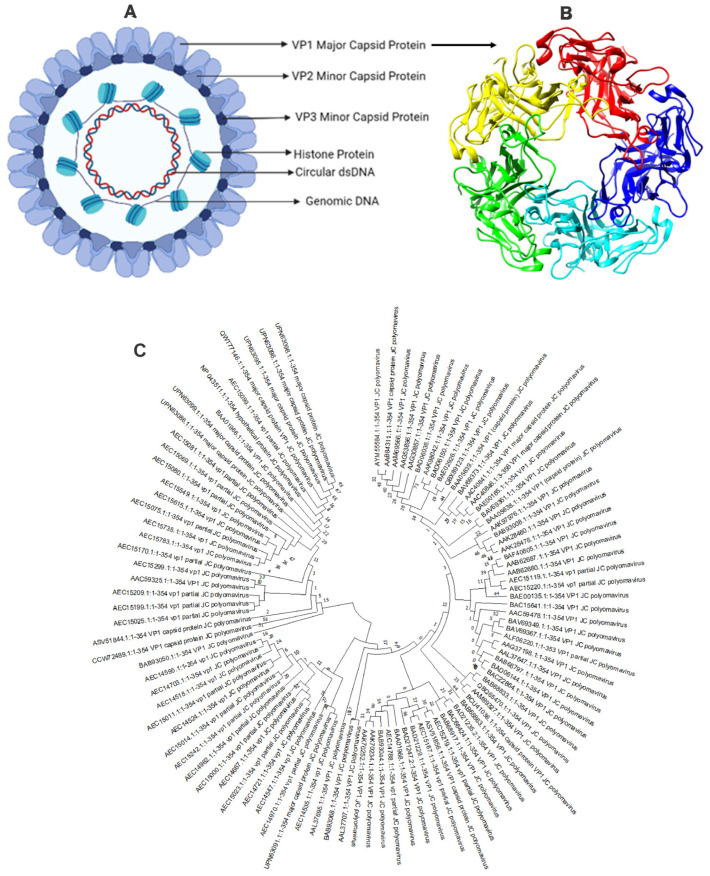
Structure of JC polyomavirus (**A**) JC polyomavirus is shown schematically, and (**B**) the primary capsid protein of JC polyomavirus is shown in a three-dimensional view using 3NXG.pdb through PyMol Software. The primary capsid protein VP1, which makes up 72 pentamers in the JC virus capsid, is present alongside lesser capsid proteins VP2 and VP3; (**C**) phylogenetic analysis of VP1 major coat protein of JC polyomavirus. The rate of diversification is lower in VP1.

**Figure 3 vaccines-11-01182-f003:**
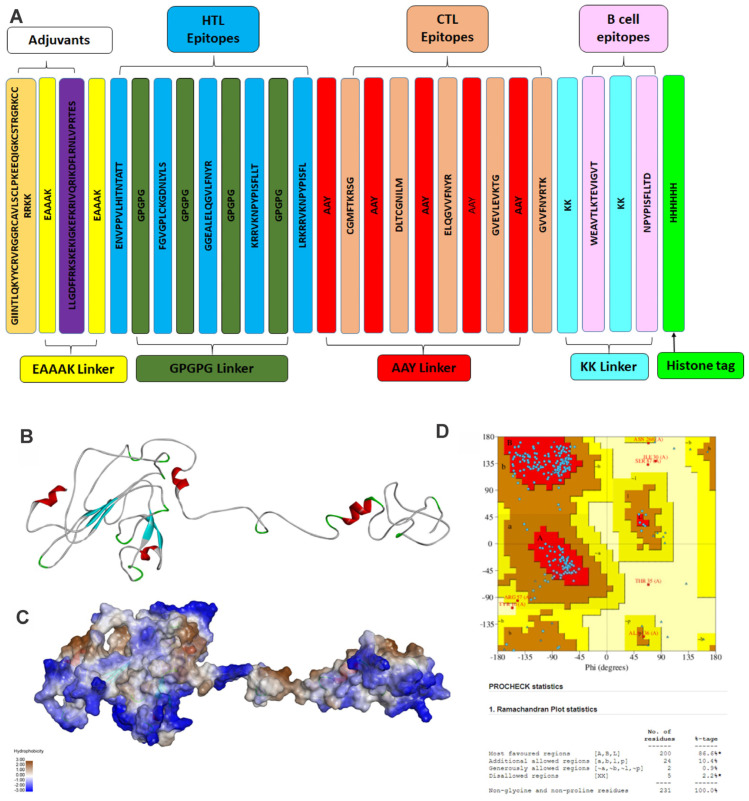
Structure of vaccine construct. (**A**) The linear vaccine construct with CTL, HTL, and B cell epitopes depicted in sea green, pink, and green boxes, respectively. Here, the adjuvant was linked using an EAAAK linker (yellow), while the epitopes were linked using aGPGPG linkers (green). The linker design consists of Human β defensin 3 –EAAAK-LL-37–EAAAK-Class 2 epitopes (GPGPG)-MHC Class 1 epitope (AAY)-B cell epitopes-KK-HHHHHH; (**B**) the three-dimensional structure of the vaccine construct, (**C**) with hydrophobic surface area, and (**D**) stereochemical quality of the vaccine construct using a Ramachandran plot. Here, A, B, and L show the residues in the most favored region while a, b, p, and l show the residues in the additional allowed regions.

**Figure 4 vaccines-11-01182-f004:**
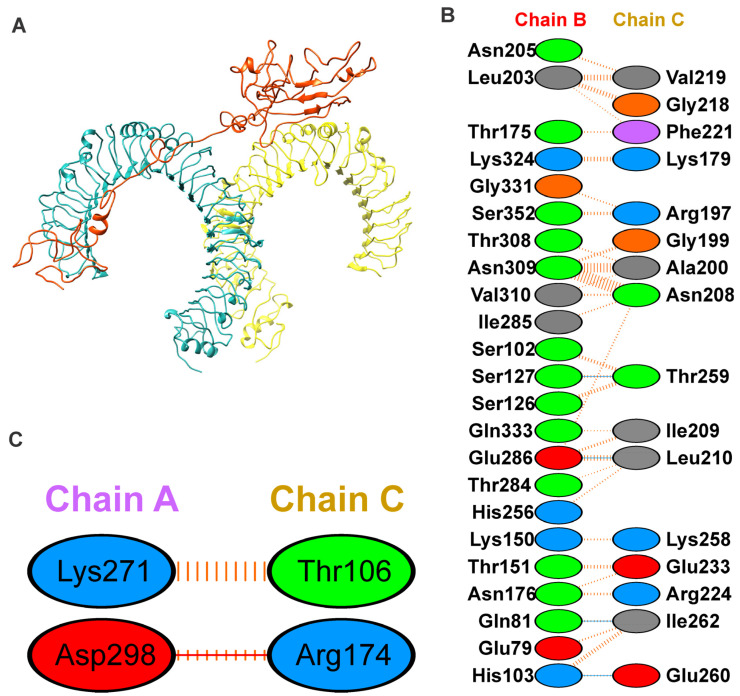
Binding mode and interaction of vaccine with TLR4 using docking. (**A**) The docked complex of the vaccine construct (orange) with Toll-like receptor 4 (chain A, cyan and chain B, yellow) using molecular docking; (**B**,**C**) interaction network of the vaccine construct with chain B and A of TLR4, respectively; hydrogen bonding interactions are shown in blue, non-bonded interactions and slat bridge are shown in orange. The positive, negative, neutral, aliphatic and aromatic charged residues are shown in blue, red, green, grey and purple color, respectively.

**Figure 5 vaccines-11-01182-f005:**
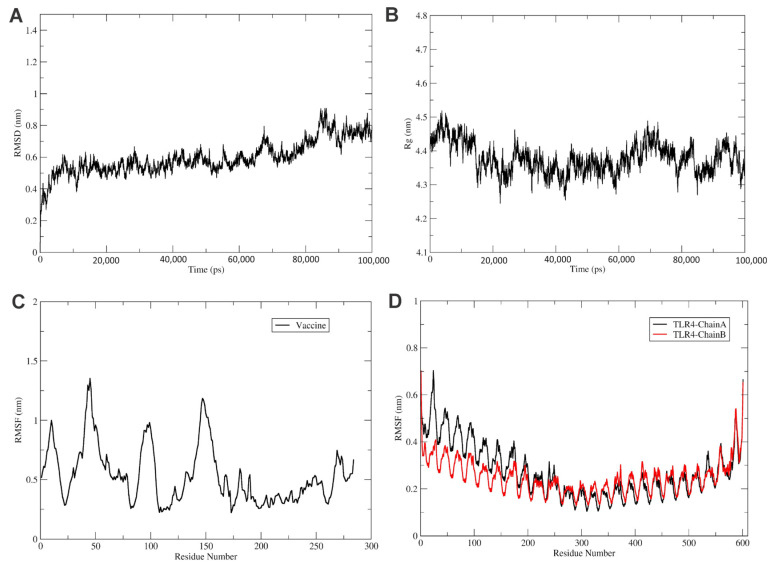
Analysis of MD simulation of TLR4 vaccine. (**A**) The RMSD of the TLR4-Vaccine complex for 100 ns. RMSD revealed that the vaccine and TLR4 receptor forms a stable complex. Similarly, (**B**) shows the radius of the gyration plot, which reveals that the complex is stable. (**C**) The RMSF of the vaccine, and (**D**) the RMSF of TLR4 receptor chains A (black) and B (red) for 100 ns time steps.

**Figure 6 vaccines-11-01182-f006:**
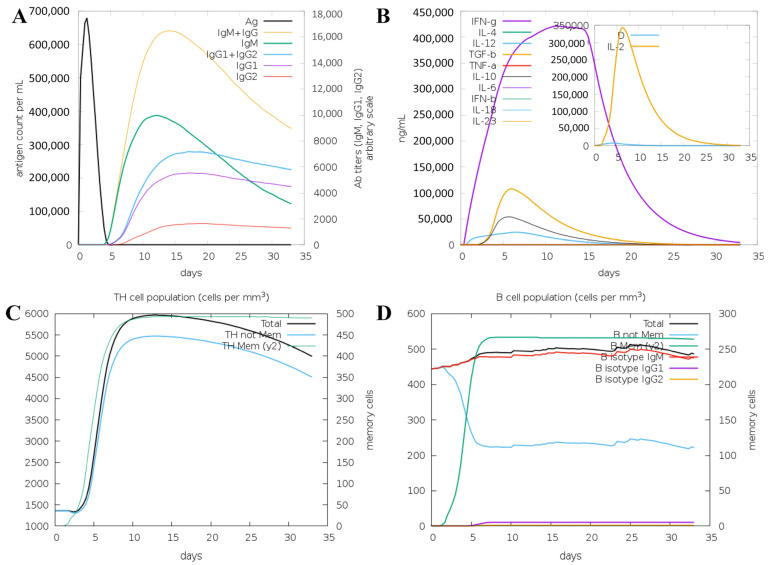
The immune response against VP1 vaccine protein as an antigen using the C-IMMSIM server. Simulations are presented after the next three injections at steps 1, 84, and 168. (**A**) The antigen and immunoglobins, and (**B**) the cytokine production. (**C**) The TH cell population, and (**D**) the B cell population.

**Table 1 vaccines-11-01182-t001:** List of selected B cell epitopes.

Peptide	Length	Antigenicity	Allergenicity	Toxicity
WEAVTLKTEVIGVT	14	Antigenic	Non-Allergen	Non-Toxin
NPYPISFLLTD	11	Antigenic	Non-Allergen	Non-Toxic

**Table 2 vaccines-11-01182-t002:** List of selected HTL cell epitopes.

Peptide	Length	Antigenicity	Allergenicity	Toxicity
ENVPPVLHINTATT	15	Antigenic	Non-Allergen	Non-Toxin
KRRVKNPYISFLLT	15	Antigenic	Non-Allergen	Non-Toxin
LRKRRVKNPYISFL	15	Antigenic	Non-Allergen	Non-Toxin
FGVGPLCKGDNLYLS	15	Antigenic	Non-Allergen	Non-Toxin
GGEALELQGVLFNYR	15	Antigenic	Non-Allergen	Non-Toxin

**Table 3 vaccines-11-01182-t003:** List of selected CTL cell epitopes.

Peptide	Length	Antigenicity	Allergenicity	Toxicity
CGMFTKRSG	9	Antigenic	Non-Allergen	Non-Toxin
DLTCGNILM	9	Antigenic	Non-Allergen	Non-Toxin
ELQGVVFNYR	10	Antigenic	Non-Allergen	Non-Toxin
GVEVLEVKTG	10	Antigenic	Non-Allergen	Non-Toxin
GVVFNYRTK	9	Antigenic	Non-Allergen	Non-Toxin

## Data Availability

All relevant data are within the paper. Further inquiries can be directed to the corresponding author.

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
