# Peer review of "Designing a Multi-Epitope Subunit Vaccine against VP1 Major Coat Protein of JC Polyomavirus"

_vaccines, 2023, doi:10.3390/vaccines11071182_

Round 1

Reviewer 1 Report (Previous Reviewer 2)

It can accept in the current state.

Author Response

Comment 1. It can accept in the current state.

Response to comment 1: We are thankful to the reviewer’s time and efforts in evaluating our manuscript and comments on our manuscript to accept in its current state. 

Reviewer 2 Report (New Reviewer)

The authors try to aim for a critical issue in vaccine development against diverse diseases. However, it is an overly simplistic approach to be published in a journal such as Vaccine-MDPI, where in my consideration, if maintaining the in-silico approach may need at least an in vitro validation.  No novelty presented in the bioinformatics workflow should lead to validation and it's the case in the presented study.

Author Response

Comment 1: The authors try to aim for a critical issue in vaccine development against diverse diseases. However, it is an overly simplistic approach to be published in a journal such as Vaccine-MDPI, where in my consideration, if maintaining the in-silico approach may need at least an in vitro validation.  No novelty presented in the bioinformatics workflow should lead to validation and it's the case in the presented study.

Response to comment: We are thankful to you for your comment and feedback on our study. We appreciate your perspective regarding the complexity of the approach and the importance of validation in vaccine development research. In our study, we aimed to address a critical issue in vaccine development by utilizing a bioinformatics workflow. It is worth noting that in-silico analyses play a valuable role in identifying potential targets, predicting antigenicity, and guiding experimental design in vaccine development. The primary focus of our study was to propose a novel bioinformatics workflow and highlight its potential utility in the early stages of vaccine design. We intended to provide a foundation for future research and stimulate discussions within the scientific community regarding this approach. We appreciate your concern regarding the limitations of an in-silico approach and the importance of in vitro validation to support our findings. However, at present, we face constraints in conducting in vitro experiments due to a lack of facilities. Nevertheless, we remain committed to advancing vaccine design and development and are actively seeking funding based on our existing data to support future research endeavors.

Our study contributes to the existing body of knowledge by proposing a systematic approach and highlighting the potential benefits of integrating bioinformatics tools in vaccine development. Our study can be a starting point for more comprehensive investigations in this field. Thank you once again for your valuable input.

Reviewer 3 Report (New Reviewer)

Thank you for the opportunity to review the manuscript entitled " Designing a Multi-Epitope Subunit Vaccine against VP1 Major Coat Protein of JC Polyomavirus ". This study employs immunoinformatics and molecular dynamics simulations to design a multi-epitope subunit vaccine targeting JC polyomavirus (JCPyV), which affects over 80% of the human population and can lead to fatal progressive multifocal encephalopathy (PML). The designed vaccine aims to stimulate immune responses against JCPyV, and molecular docking shows a stable complex formation with TLR4.The introduction is well-written, providing a clear background on PML, JCV, and the importance of the primary capsid protein VP1 in JCV attachment and infection. The context of vaccination in light of the SARS-CoV-2 outbreak and immunoinformatics approaches is also well-explained. The results section presents a comprehensive account of the findings, including the antigenicity, stability, and other relevant properties of the predicted epitopes and vaccine construct. The discussion section emphasizes the challenges associated with PML and JCPyV infections and the importance of developing an effective vaccine. The authors describe their efforts to design a vaccine construct using immunoinformatics and molecular modeling techniques. I have conscientiously evaluated your work and would like to respectfully offer some comments and clarifications for improvement.

Major comments:

1.       Provide a more detailed explanation of the residues involved in the TLR4-Vaccine complex interactions in the molecular docking subsection.

2.       Elaborate on the implications of the observed immune response in the immune system simulation.

3.       Provide a more in-depth discussion of the implications of the findings for the development of a JCPyV vaccine and how the proposed vaccine construct might address the challenges associated with PML and JCPyV infections.

Author Response

General Comment: Thank you for the opportunity to review the manuscript entitled " Designing a Multi-Epitope Subunit Vaccine against VP1 Major Coat Protein of JC Polyomavirus ". This study employs immunoinformatics and molecular dynamics simulations to design a multi-epitope subunit vaccine targeting JC polyomavirus (JCPyV), which affects over 80% of the human population and can lead to fatal progressive multifocal encephalopathy (PML). The designed vaccine aims to stimulate immune responses against JCPyV, and molecular docking shows a stable complex formation with TLR4. The introduction is well-written, providing a clear background on PML, JCV, and the importance of the primary capsid protein VP1 in JCV attachment and infection. The context of vaccination in light of the SARS-CoV-2 outbreak and immunoinformatics approaches is also well-explained. The results section presents a comprehensive account of the findings, including the antigenicity, stability, and other relevant properties of the predicted epitopes and vaccine construct. The discussion section emphasizes the challenges associated with PML and JCPyV infections and the importance of developing an effective vaccine. The authors describe their efforts to design a vaccine construct using immunoinformatics and molecular modeling techniques. I have conscientiously evaluated your work and would like to respectfully offer some comments and clarifications for improvement.

Response to general comment: Thank you for taking the time to review our manuscript. We appreciate your positive feedback and valuable comments on our study. We sincerely appreciate your thoughtful evaluation of our work and the opportunity to address your comments. Your feedback will significantly contribute to improving the manuscript. We will carefully consider your suggestions and make the necessary revisions to enhance the quality and clarity of our study. Thank you again for your valuable input.

Comment 1: Provide a more detailed explanation of the residues involved in the TLR4-Vaccine complex interactions in the molecular docking subsection.

Response to comment: Thank you for the suggestion. We have included the detailed discussion on the residues involved in the TLR4 and vaccine interaction in the result and discussion section. (Line: 270-274). 

Comment 2: Elaborate on the implications of the observed immune response in the immune system simulation.

Response to comment:  We thank the reviewer for suggestions for to elaborate the immune response in the immune system simulation. In the revised manuscript, we have disucss the immune resonse of the vaccine construct. These assessments contribute to ensuring the immunogenicity and stability of the vaccine construct, which are crucial factors for successful vaccine development. (Line: 312-322)

Comment 3: Provide a more in-depth discussion of the implications of the findings for the development of a JCPyV vaccine and how the proposed vaccine construct might address the challenges associated with PML and JCPyV infections.

Response to comment: We are thankful to the reviewer for their valuable comment. In our revised manuscript, we have added a comprehensive discussion on the implications of our findings for the development of a JC polyoma virus (JCPyV) vaccine. We elaborated that how our proposed vaccine construct addresses the challenges linked to progressive multifocal leukoencephalopathy (PML) and JCPyV infections. Additionally, we discussed the advantages of our multi-epitope subunit vaccine, including enhanced immunogenicity and reduced adverse reactions compared to traditional vaccines. (Line: 378-387)

Round 2

Reviewer 2 Report (New Reviewer)

Besides the improvement of some parts of the manuscript, no substantial enhancement of the scientific quality is noticed in the present version. I do insist that for publication in a journal such as Vaccine, a in silico study must present an innovation in the field or at least an in vitro validation of the analysis to be further considered.

Author Response

Response to Reviewer #1

Comment: Besides the improvement of some parts of the manuscript, no substantial enhancement of the scientific quality is noticed in the present version. I do insist that for publication in a journal such as Vaccine, a in silico study must present an innovation in the field or at least an in vitro validation of the analysis to be further considered.

Response to comment: Thank you for your thorough review of our manuscript. We sincerely appreciate the time and effort you have dedicated to providing us with valuable feedback. We have carefully considered your comments regarding the scientific quality of our study. We understand your concern regarding the level of scientific enhancement in the current version of the manuscript and we substantially proof read the manuscript and enhance the quality, also we used standared methods in the present study. We would like to clarify that our study primarily focused on an in silico analysis, which involved the validation of our findings using docking and explicit molecular dynamics simulations. While we recognize that computational modeling alone may not provide all the necessary scientific evidence, we believe that in silico studies can still contribute valuable insights and predictions in certain contexts. However, we acknowledge the need for additional scientific evidence beyond computational modeling, such as in vitro experiments, to further strengthen the validity of our findings. Unfortunately, due to resource constraints, we currently face challenges in conducting in vitro experiments as we lack the necessary facilities. Nevertheless, we remain committed to advancing vaccine design and development. We are actively seeking funding based on our existing data to support future research endeavors, including in vitro validation. We believe that our study contributes to the existing body of knowledge by proposing a systematic approach and highlighting the potential benefits of integrating bioinformatics tools in vaccine development. It can serve as a starting point for more comprehensive investigations in this field.

This manuscript is a resubmission of an earlier submission. The following is a list of the peer review reports and author responses from that submission.

Round 1

Reviewer 1 Report

1. The quality of Figure 4 should be improved.

2. Where the Figure 6 as mentioned ion page 11?

3. The reasionable of the docking structure of designed vacine  to TLR-4 should be further evaluated. Is there any experimental structure like VP1 bind to TLR-4, which can be used to verifiy the predicted binding mode. In addition, the docking pose can be validated by using the energy funnel as described in Chem Biol Drug Des. 2021, 98(1):1-18.

4. To combat viral infection, desined proteins like vaccine in this work may provided important role. Recently, a database refer to synthetic binding proteins (Nucleic Acids Res. 2022, 50(D1):D560-D570.) has been released, which should be introduced in the introduction section.

Author Response

Response to comments:

Comment 1. The quality of Figure 4 should be improved.

Response to comment 1: Thank you for your valuable suggestion. We have taken your feedback into consideration and have made improvements to the quality of Figure 4 in the revised version of our manuscript. We appreciate your input, which has helped enhance the visual representation of our research findings and better convey our scientific message.

Comment 2. Where the Figure 6 as mentioned on page 11?

Response to comment 2: Thank you for your helpful suggestion. We have taken your feedback into account and made the necessary changes to the revised version of our manuscript. Specifically, we have updated the figure number to accurately reflect the content of the figure. We appreciate your input, which has helped to improve the quality and accuracy of our research publication.

Comment 3. The reasionable of the docking structure of designed vacine  to TLR-4 should be further evaluated. Is there any experimental structure like VP1 bind to TLR-4, which can be used to verifiy the predicted binding mode. In addition, the docking pose can be validated by using the energy funnel as described in Chem Biol Drug Des. 2021, 98(1):1-18.

Response to comment 3: We are extremely grateful for the valuable suggestions provided by the reviewer, which have helped to enhance the scientific rigor of our manuscript. As per the suggestion, we have included an interaction analysis of the docked complex in the revised version of our manuscript. To perform this analysis, we utilized the PDBsum database (https://www.ebi.ac.uk/thornton-srv/databases/pdbsum/Generate.html) to examine the interaction network of the vaccine-TLR4 receptor complex.

Additionally, while structural information about VP1 and TLR4 binding has not yet been studied, an earlier study has demonstrated that recognition of Mouse Polyomavirus (MPyV) by Toll-like receptor 4 (TLR4) induces the production of interleukin 6 and other cytokines without inhibiting virus multiplication (Janovec et al., Cancers 2021, 13, 2076).

Regarding the docking results, as per suggestion, validating them using the Rosetta energy funnel. While the Rosetta docking mechanism is indeed necessary for the Rosetta energy funnel to function, we opted to use PatchDock and FireDock servers in this instance to obtain the least-energy conformation of TLR4 and the vaccine. This approach is consistent with our earlier work (Sharma et al., Immunobiology, 2021 Mar;226(2):152053. doi: 10.1016/j.imbio.2021.152053) (Reference 15-18) and allowed us to rigorously assess the docking results.

We appreciate the reviewer's thoughtful feedback and their contribution to the refinement of our manuscript.

Comment 4. To combat viral infection, designed proteins like vaccine in this work may provided important role. Recently, a database refer to synthetic binding proteins (Nucleic Acids Res. 2022, 50(D1):D560-D570.) has been released, which should be introduced in the introduction section.

Response to comment 4: We are deeply appreciative of the reviewer's valuable suggestions, which have greatly strengthened the scientific content of our manuscript. As per their suggestion, we have included the recommended references in the revised version of our paper (Ref 19). We believe that this has significantly enhanced the quality and comprehensiveness of our research findings, and we are grateful for the reviewer's insightful feedback.

Reviewer 2 Report

JC polyomavirus (JCPyV) is the causative agent of a fatal central nervous system demyelinating disease known as progressive multifocal leukoencephalopathy. Virus protein 1 (VP1) is often targeted for antiviral drug and vaccine development. This study implied immunoinformatics and molecular dynamics simulation-based approaches to design a multi-epitope subunit vaccine targeting JCPyV. However, this article is focused primarily on using simulation studies, lacks the laboratory the evidence, unable to provide the reliable basis for the actual effects of vaccination.

1. Lacks the laboratory the evidence.

2. English should certainly be improved.

3. The picture editing is chaotic and lacks the correct order. For example, Figure 2 appears after Figure 3.

4. Safety evaluation test of vaccine should be added.

5. Figure 3 contains the results that should be placed in the results section.

6. The Discussion could be fuller, perhaps combining the author’s earlier work with the present results to discuss how the research should develop.

Author Response

General comment: JC polyomavirus (JCPyV) is the causative agent of a fatal central nervous system demyelinating disease known as progressive multifocal leukoencephalopathy. Virus protein 1 (VP1) is often targeted for antiviral drug and vaccine development. This study implied immunoinformatics and molecular dynamics simulation-based approaches to design a multi-epitope subunit vaccine targeting JCPyV. However, this article is focused primarily on using simulation study lacks laboratory evidence, unable to provide a reliable basis for the actual effects of vaccination.

Response to general comment: We would like to express our sincere gratitude to the reviewer for dedicating their time and expertise to evaluate our manuscript. Their insightful comments and valuable suggestions have greatly contributed to the improvement of our research work, and we appreciate their efforts in helping us to strengthen our scientific findings. Once again, we extend our heartfelt thanks to the reviewer for their invaluable contribution to our manuscript.

Comment 1. Lacks the laboratory the evidence.

Response to comment 1: "Thank you for taking the time to review our manuscript. We are grateful for your constructive feedback and fully acknowledge the significance of laboratory evidence to validate our proposed hypothesis. At present, we are in the process of prediction and designing of vaccine, and we are committed to incorporating these experimental results into the next phase of our study. We sincerely appreciate your valuable inputs, which have significantly contributed to the refinement and overall quality of our research."

Comment 2. English should certainly be improved.

Response to comment 2: Thank you for your feedback on our manuscript. Based on your suggestion, we have carefully reviewed and revised the language in the manuscript to ensure clarity and accuracy. We have made significant improvements to the English language, with the aim of making the manuscript more accessible to a wider audience. We appreciate your input and believe that these revisions have strengthened the overall quality of the manuscript. Thank you for helping us to improve our work.

Comment 3. The picture editing is chaotic and lacks the correct order. For example, Figure 2 appears after Figure 3.

Response to comment 3: Thank you for taking the time to provide feedback on our manuscript. We greatly value your insights and have taken your comments into careful consideration. We are pleased to inform you that we have made significant improvements to the image quality in the revised manuscript, thanks to your valuable suggestions. We believe that these enhancements will greatly enhance the reader's experience and overall understanding of our work. Once again, we appreciate your constructive feedback and are grateful for the opportunity to address it in our revised manuscript.

Reviewers comment 4. Safety evaluation test of vaccine should be added.

Response to comment 4: Thank you to the reviewer for your invaluable feedback on our manuscript. We greatly appreciate your insights and suggestions. While it may not be possible to include a thorough safety evaluation of the proposed vaccine in this current manuscript, we are pleased to inform you that we have carefully considered your recommendations and plan to incorporate a comprehensive safety evaluation in the next parts of our study. We are confident that these updates will significantly enhance the scientific rigor and translational impact of our work. Once again, we would like to express our sincere gratitude for your contribution to our research, and we look forward to your continued feedback.

Reviewers comment 5. Figure 3 contains the results that should be placed in the results section.

Response to comment 5: Thank you to the reviewer for your valuable suggestion. We greatly appreciate your careful review of our manuscript. We are pleased to inform you that we have taken your feedback into account and have updated the figure number accordingly in the revised version of the manuscript. We believe that this revision will significantly improve the clarity and accessibility of our work.

Reviewers comment 6. The Discussion could be fuller, perhaps combining the author’s earlier work with the present results to discuss how the research should develop.

Response to comment 6: Thank you for your valuable suggestion. We greatly appreciate your feedback and have taken it into consideration. We are pleased to inform you that we have now included our earlier work in conjunction with our present introduction and discussion (Ref 15-18), in order to provide a more comprehensive and cohesive analysis of the topic at hand. We hope that this addition will further enhance the quality and value of our study.

Round 2

Reviewer 1 Report

The manuscript has been significantly revised according the reviewers comments. For comment 4, the ref.19 in the revised manuscript was not suitable.

Reviewer 2 Report

This article is focused primarily on using simulation study lacks laboratory evidence, unable to provide a reliable basis for the actual effects of vaccination.